# Low-Level Groundwater Atrazine in High Atrazine Usage Nebraska Counties: Likely Effects of Excessive Groundwater Abstraction

**DOI:** 10.3390/ijerph182413241

**Published:** 2021-12-15

**Authors:** Moses New-Aaron, Olufemi Abimbola, Raheleh Mohammadi, Oluwaseun Famojuro, Zaeema Naveed, Azar Abadi, Jesse E. Bell, Shannon Bartelt-Hunt, Eleanor G. Rogan

**Affiliations:** 1Department of Environmental Health, Occupational Health and Toxicology, University of Nebraska Medical Center, Omaha, NE 68198, USA; azar.abadi@unmc.edu (A.A.); jesse.bell@unmc.edu (J.E.B.); egrogan@unmc.edu (E.G.R.); 2Department of Biological Systems Engineering, University of Nebraska-Lincoln, Lincoln, NE 68583-0726, USA; 3Department of Epidemiology, University of Nebraska Medical Center, Omaha, NE 68198, USA; raheleh.mohammadi@unmc.edu (R.M.); oluwaseun.famojuro@unmc.edu (O.F.); 4School of Population and Public Health, The University of British Columbia, Vancouver, BC V6T 1Z3, Canada; zaeema_naveed@ubc.edu; 5Department of Civil and Environmental Engineering, University of Nebraska-Lincoln, Omaha, NE 68182-0178, USA; sbartelt@unl.edu

**Keywords:** groundwater, atrazine, abstraction, cancers, climate, Nebraska

## Abstract

Recent studies observed a correlation between estrogen-related cancers and groundwater atrazine in eastern Nebraska counties. However, the mechanisms of human exposure to atrazine are unclear because low groundwater atrazine concentration was observed in counties with high cancer incidence despite having the highest atrazine usage. We studied groundwater atrazine fate in high atrazine usage Nebraska counties. Data were collected from Quality Assessed Agrichemical Contaminant Nebraska Groundwater, Parameter–Elevation Regressions on Independent Slopes Model (PRISM), and water use databases. Descriptive statistics and cluster analysis were performed. Domestic wells (59%) were the predominant well type. Groundwater atrazine was affected by well depth. Clusters consisting of wells with low atrazine were characterized by excessive groundwater abstraction, reduced precipitation, high population, discharge areas, and metropolitan counties. Hence, low groundwater atrazine may be due to excessive groundwater abstraction accompanied by atrazine. Human exposure to atrazine in abstracted groundwater may be higher than the estimated amount in groundwater.

## 1. Introduction

Approximately 115 million people in the U.S. rely on groundwater as drinking water [1], and 80–85% of Nebraskans receive their drinking water from groundwater [2,3]. Despite the importance of this water source, there are many unresolved issues about its quality. Water quality standards of private wells are not regulated under the Safe Drinking Water Act; however, Nebraska Departments of Agriculture (N.D.A.) and Environmental Quality initiated a project in 1996 to create a data repository for groundwater that would allow the assessment of groundwater pesticides obtained at different periods for different purposes [3]. This data repository is called the Quality-Assessed Agrichemical Contaminant Data for Nebraska. We recently explored this database to identify the different pesticides in Nebraska groundwater and their likely health implications. We observed clusters of breast and prostate cancers in counties with positive atrazine groundwater [4].

Moreover, only low-level atrazine was detected in most wells of counties with higher cancer incidence despite the high atrazine usage in these counties. The discordance between atrazine usage and groundwater atrazine concentration raises a critical question, especially when no known groundwater atrazine depletion intervention is in place in these counties. To address this conundrum, this study explored groundwater atrazine fate to account for missing groundwater atrazine residue after land application. Understanding atrazine fate will help uncover the exposure mechanisms of groundwater atrazine in counties of high atrazine usage.

Meanwhile, atrazine and its metabolites were not the only detected agrichemical in Nebraska groundwater; other agrichemicals, such as nitrate, glyphosate, acetochlor, and alachlor, were also detected. This may be due to the co-usage of atrazine with other agrichemicals. In fact, glyphosate usage is as high as atrazine usage in Nebraska [5]. While glyphosate usage did not become widespread until recently, other herbicides such as alachlor are as old as atrazine [6,7]. Despite atrazine co-usage with other herbicides, only atrazine and its metabolites persist longer in groundwater [8,9]. This fact underscores why this study is focused on atrazine in Nebraska groundwater. Atrazine persistence may be linked to high application rates; other factors such as well structures, groundwater abstraction rates, and climatic changes [10] may play crucial roles in groundwater atrazine’s fate and human exposure.

To effectively understand the health implications of atrazine, knowledge of atrazine’s fate in groundwater is vital. Moreover, designing studies in line with Bradford Hills criteria for evaluating the cause-and-effect relationship between atrazine exposure and disease outcomes will become apparent with the in-depth understanding of atrazine fate in groundwater.

In addition, atrazine metabolites are often ignored when exploring atrazine toxicity, even though these metabolites may have significant pathological implications [11]. In fact, previous studies have characterized the toxicity of atrazine metabolites, Deethylatrazine (D.E.A.) and Deisopropylatrazine (D.I.A.), as endocrine disruptors in humans and among aquatic organisms [12,13]. Therefore, it should not be assumed that atrazine degradation results in the detoxification of atrazine. Thus, exploring the fate of toxic groundwater atrazine metabolites in an agricultural setting with high atrazine usage will contribute to this body of knowledge.

To provide clean and safe water, atrazine and other agrichemicals are frequently measured in groundwater. Since these measurements only detect low-level atrazine even in high atrazine usage counties, it leaves one to wonder whether the atrazine measurements represent actual groundwater atrazine deposition. It could be that atrazine concentration is underestimated. Given this, we aimed to determine the groundwater atrazine fate of selected Nebraska counties with high atrazine usage. Nebraska is a good subject for this study because it is one of the agriculturally intensive “corn belts” of the United States. The objective of this study was to uncover the potential reasons for the frequently observed low-level groundwater atrazine in eastern Nebraska counties, which are characterized by high atrazine usage.

## 2. Materials and Methods

Data used for this county-level study were obtained from three data sources: Quality Assessed Agrichemical Contaminant Nebraska Groundwater Database; Parameter–Elevation Regressions on Independent Slopes Model (PRISM) as weather data [14,15]; and water use data obtained from United States Geographical Survey (USGS).

Counties with high atrazine usage (>28.73 kg/mi^2^), as indicated by the National Water-Quality Assessment (NAWQA) Project, USGS (1992–2017), were included in this study. Based on this, 33 counties in the eastern Nebraska District (Burt, Butler, Cass, Cedar, Colfax, Cuming, Dakota, Dixon, Dodge, Douglas, Fillmore, Gage, Jefferson, Johnson, Lancaster, Lincoln, Madison, Nemaha, Otoe, Pawnee, Pierce, Platte, Polk, Richardson, Saline, Sarpy, Saunders, Seward, Stanton, Thayer, Washington, Wayne, York) were eligible for this study, Figure 1a,b. The findings from our recent study, which observed a potential correlation between atrazine and estrogen-related cancers (ERC) in eastern Nebraska, further justified the selection of the study area [4].

While USGS pesticide usage data began in 1992, the timeframe for this study was between 1995 and 2014 due to data availability for atrazine-contaminated groundwater. The Quality Assessed Agrichemical Contaminant Nebraska Groundwater Database was queried for atrazine and its metabolites (D.E.A., D.I.A., and hydroxyatrazine) for 1995–2014. In addition to the concentration of atrazine and its metabolites in parts per billion (ppb), other variables such as well types and well depths (in feet) were also obtained. Methods used for measuring atrazine, D.E.A., and D.I.A. concentrations in the water supply wells were described elsewhere [16]. The water supply wells selected for this study were wells in eastern Nebraska counties with high atrazine usage. Numerous wells were measured for atrazine in each county, and each well was measured multiple times during the study period.

Since groundwater atrazine fate in saturated and unsaturated aquifers is greatly impacted by environmental factors, such as precipitation and soil temperature across a range of soil profiles and over time, there was a need to incorporate some of these factors into the analysis. However, since the network of land-based weather stations may lack the capacity to adequately capture the spatial variability of weather variables across the counties mentioned above, the PRISM weather dataset was used as an alternative in this study. The PRISM is a high-resolution weather dataset based on a spatial resolution of 4 km. Daily time series data for precipitation and mean air temperature were extracted from 1995 to 2014. Although soil temperature would be more critical to the kinetics of atrazine in groundwater than the air temperature, the lack of direct measurements of soil temperature resulted in the use of annual mean air temperature as a proxy for the soil temperature at depths where groundwater wells would be screened. This is valid because there is a strong relationship between mean air temperature and soil temperature due to the exchange processes between them [17,18]. Groundwater temperature is usually equal to the annual mean air temperature above the ground, and it generally fluctuates narrowly (based on depth) around this mean temperature year round.

USGS via the web interface of the National Water Information System provides water usage data for different surface or groundwater types. This database was queried for water usage between 1995 and 2010 because 2014 data was unavailable. While annual groundwater usage was unavailable, the report was available every five years for the designated counties. Since domestic well usage may be a better predictor of human exposure to groundwater atrazine, the total number of people using self-supplied domestic fresh groundwater and the amount of Million gallons per day (Mgal/d) of fresh domestic groundwater withdrawn in the selected 33 counties of eastern Nebraska were obtained.

### Statistical Analysis

Variables included in the analysis were either categorical or continuous variables. Descriptive analyses were performed on the categorical variables (groundwater or well type). Meanwhile, the time (in years) of sampling groundwater atrazine, D.E.A., and D.I.A., which was initially a count variable, was categorized by five year intervals. Continuous variables were atrazine, D.E.A., D.I.A. concentrations (ppb), well depth (in feet), precipitation (in millimeters), and annual mean air temperature (in degree Celsius). Descriptive statistics for these variables included mean, standard deviation, minimum and maximum values. Given the longitudinal design of this study, we used scatter plots to examine the correlations between atrazine, D.I.A., D.E.A. concentrations (ppb), and time (years).

A cluster analysis was performed to examine the effects of well depth on groundwater atrazine concentration. Similarity for each cluster was based on the negative squared Euclidean distance of both standardized atrazine and well depth, and the shared value was 20% quantile of their similarities.

All analyses were performed on SASv9.4 (S.A.S. Institute Inc. 2013. Cary, NC, USA), and plots were made on Microsoft Excel 2016 and Prism GraphPad Prism v7.03 software (GraphPad, La Jolla, CA, USA).

## 3. Results

### 3.1. Descriptive Statistics of Sampled Wells, Hydrometeorological Characteristics, and Groundwater Utilization in the Eastern Nebraska Counties

This study included six well types (commercial, domestic, irrigation, public, monitoring, and livestock wells). Domestic wells were the most represented well-type, accounting for 59% of the study wells (Figure 2a). Furthermore, irrigation (180 ft) and domestic wells (120 ft) were the deepest of all well types in the study location (Figure 2b).

The average values of atrazine, D.E.A., and D.I.A. during the entire study period (1995–2014) for all the counties were 0.17, 0.015, and 0.073 ppb, respectively. However, no value was obtained for hydroxyatrazine, another atrazine metabolite, during this period. Furthermore, the overall average well depth (129.94 ft) is similar to the average depth of domestic wells, confirming the high prevalence of domestic well types among the study wells. Interestingly, the average withdrawals of domestic groundwater were 0.90 million gallons per day (Mgal/day), and these supplied an average of 7100 people in the selected counties of eastern Nebraska based on 2010 data (Table 1).

### 3.2. Atrazine Concentration by Well Depth in Eastern Nebraska Counties

Atrazine, D.E.A., and D.I.A. concentration were higher in shallow wells (Figure 3a–c).

### 3.3. Depletion of Atrazine and Its Metabolites with Time in Eastern Nebraska Counties

In Figure 4a, a time-dependent groundwater atrazine decline was observed despite continuous atrazine usage during the same period. This corresponded to a decrease in groundwater D.E.A. and D.I.A. (Figure 4b,c).

### 3.4. Precipitation and Temperature trend in Eastern Nebraska Counties

Precipitation and temperature trends between 1995 and 2014 were characterized in Figure 5a,b, respectively. The observed counties seem to record lower precipitation in 1995, 2000, and 2012, Figure 5a.

### 3.5. Seasonal Variation of Atrazine and Its Metabolites in Eastern Nebraska Counites

Figure 6a–c shows that the average groundwater atrazine, D.E.A., and D.I.A. concentration by month was consistently higher during Nebraska’s winter and early springs (December-March). Moreover, the highest precipitation was recorded in May-June (Figure 6d), and the highest mean daily temperature was reported in July-August (Figure 6e).

### 3.6. Characterization of Groundwater Atrazine Depletion in Eastern Nebraska Counties

Given that irrigation and domestic water wells were the deepest wells in the study area and atrazine depletion was observed in deeper wells, we clustered the data based on well depth and re-evaluated the effects of other factors on atrazine depletion. Seven different clusters of counties were observed. While wells in counties of clusters 3, 4, and 5 had low-level atrazine regardless of well depths, wells found in cluster 7 had high atrazine concentration, Figure 7a. To exclude the effect of well depth, additional analysis was performed, comparing two different clusters with similar well depth but different groundwater atrazine concentrations, Table 2. The two clusters eligible for this comparison were cluster 5 (low atrazine) and cluster 7 (high atrazine). Cluster 5 contains wells mostly in groundwater discharge areas, while cluster 7 contains wells predominantly in groundwater recharge areas. Groundwater discharge areas and recharge areas are areas where groundwater flow has an upward and a downward flow component, respectively. The discharge areas of a regional groundwater system are located downstream of a river basin, while the recharge areas of a regional system occupy the upstream water divide of the river basin. For a local groundwater flow system, its discharge areas are at a topographic low, and its recharge areas are at an adjacent topographic high. As shown in the figure below, for cluster 5 counties, most of Burt, Dodge, and Colfax counties are in the downstream areas of Elkhorn River Basin, while Sarpy and Cass counties are in the downstream areas of both Lower Platte and Missouri River Basins.

Similarly, the northwestern part of Lincoln County is in the downstream areas of both North Platte and South Platte River Basins. Although the wells in the eastern part of Lincoln County are located in upstream of the Middle Platte River Basin, they are mainly close to Platte River, which implies that they are in topographic low (discharge areas). For cluster 7 counties, most Polk and all York counties are located in the upstream (recharge areas) of the Big Blue River Basin.

The average population that potentially used the groundwater in cluster 5 was three times more than cluster 7. Moreover, wells in cluster 5 received slightly lower precipitation than cluster 7. The average well density per unit land area is low for cluster 5 compared to cluster 7 (Figure 7b–h).

## 4. Discussion

It is rational to expect groundwater atrazine concentration in Nebraska counties with continuously high atrazine usage to be significantly elevated or at least remain constant over time. Instead, low-level groundwater atrazine is frequently observed even though no groundwater atrazine elimination process was identified in these counties. This raises some issues addressed by this study. Before delving into these critical issues, the long-term significance of this study will be reiterated. The toxic or carcinogenic effects of atrazine are common knowledge due to evidence from experimental [19,20,21,22,23] and ecological studies [4,24,25]. However, due to sparse epidemiological evidence [26,27], atrazine is often absolved of the supposed toxicity observed in experimental and ecological studies.

Meanwhile, it must be noted that only the use of individual-level atrazine exposure data would credibly predict the health outcomes associated with atrazine exposure. However, only a handful of studies are available for such designs. Hence most studies utilize county-level groundwater atrazine as exposure for potential disease outcomes. This may be downplaying the toxic effects of atrazine, since groundwater atrazine measurement reveals low-level atrazine concentration, which would interfere with identifying existing correlations between atrazine exposure and suspected pathological conditions when county-wide atrazine use data is used instead of individual atrazine exposure data. To this end, this study provided evidence for why the inclusion of groundwater atrazine measurements as explanatory variables for most models in epidemiological studies may fail to predict proposed atrazine-induced pathological conditions accurately.

The inferences drawn from this study may have direct human implications, given that approximately 60% of the sampled wells were domestic. Moreover, the domestic wells in this study were among the deepest. This is interesting and reassuring, since previous studies have observed correlations between better water quality and deeper wells [28]. Furthermore, the average atrazine amount detected in all the wells for the entire study period is significantly less than the United States Environmental Protection Agency Maximum Contaminant Level (MCL). Additionally, two primary atrazine metabolites, D.E.A. and D.I.A., were detected, suggesting atrazine degradation during the study period. While D.E.A. and D.I.A. are not the only atrazine metabolites, they were the only metabolites sufficiently detected during the study period. Hydroxyatrazine, another atrazine metabolite, was not detected, indicating dealkylation as the predominant metabolic pathway for atrazine degradation in the sampled groundwater. While abiotic pathway was previously reported for atrazine dealkylation, most atrazine dealkylation processes are attributed to biotic pathways [29]. This may suggest microbial co-contamination of the sampled groundwater [30,31,32]. Although microbial contaminant is not the focus of this study, this needs to be verified by future studies.

Atrazine depletion corresponded to D.E.A. and D.I.A. depletion in this study. Moreover, atrazine, D.I.A., and D.E.A. depletions were more apparent in the year 2000 than in other years. It is difficult to conclude any relationship between atrazine and D.I.A. or D.E.A. depletion, given that no baseline data for any of the pesticides was captured in this study. Furthermore, atrazine depletion was observed with the formation of its metabolites as time progressed. Although atrazine half-life in the sampled water supply wells may be challenging to determine, atrazine degradation to D.E.A. or D.I.A. contributes to atrazine depletion. Atrazine half-life depends on environmental factors. For example, it may range between 2 weeks and 16 weeks in surface soils. Moreover, it could be four years, or degradation may not even occur [33]. Atrazine degradation was not observed in groundwater after 77 weeks [34], and another study reported atrazine’s half-life in groundwater as 83 weeks [35]. However, this may be as short as 24 weeks in the presence of sunlight [36].

As this is an environmental observational study, environmental effects, including climatic changes, cannot be excluded from groundwater atrazine’s fate. Climatic changes were in this study described in terms of hydrometeorological factors such as precipitation and annual mean air temperature. While daily mean air temperatures observed for most of the years were in the range of extreme heat or cold, there appears to be evidence of drought in 1995, 2000, and 2012. Drought during a growing season reduces the groundwater recharge rate. Consequently, the lower recharge also reduces the leaching of atrazine to wells. Hence, the sharp atrazine, D.I.A., and D.E.A. depletion in the year with the longest drought duration may be due to a drought-induced decrease in atrazine leaching [37]. Moreover, high precipitation was observed in cluster 7, characterized by high groundwater atrazine concentration. This suggests the involvement of precipitation in the deposition of atrazine in groundwater [38].

Data used in this study provided evidence of seasonal variation of atrazine, D.E.A., and D.I.A. While May and June are the peak season for atrazine application [39,40], December, January, February, and March, which are winter/early spring seasons, were observed in this study as the months with groundwater peak atrazine, D.E.A., and D.I.A. In contrast, another study reported peak atrazine concentration in late summer and early autumn. This was attributed to rainfall [41]. Nebraska’s intense rainstorms in May and June may contribute to atrazine leaching after application. This, therefore, suggests that peak groundwater atrazine detected in the winter and early spring may result from a time lag of five to seven months required for atrazine transition from the application site to groundwater.

Moreover, other studies conducted in the Midwest have reported groundwater atrazine peaks in the winter and early spring [42], which is in congruence with the findings of this study. In addition, winter is known to slow down atrazine degradation [43]. This may partly contribute to the seasonal variation of atrazine in favor of winter and early spring.

Atrazine depletion with time was indeed observed in this study. Given the extreme climatic changes in Nebraska, one may easily attribute this to the time-dependent atrazine depletion. However, only a slight precipitation effect was observed. Instead, well depth highly predicted low-level groundwater atrazine. This finding is not novel because the associations between well depth and decreased atrazine level were previously reported [28]. This underscores the significance of the irrigation and domestic wells, the deepest wells in this study. To determine other factors beyond well depth which affects groundwater atrazine concentration, a cluster analysis was performed. The effect of well depth was excluded by comparing clusters with the same well depth but different groundwater atrazine concentrations. County clusters with low groundwater atrazine concentration had approximately three times the population supplied by domestic groundwater compared to county clusters with high-level atrazine.

Furthermore, counties with low-level atrazine are more metropolitan than counties with high-level atrazine. In contrast to the wells in high-level atrazine counties, wells in low-level atrazine counties were mostly utilization wells. This suggests that low-level groundwater atrazine in cluster 5 may be due to excessive groundwater usage [44].

In addition, cluster 5 with low-level atrazine counties (mainly in groundwater discharge areas) was characterized by slightly lower precipitation than cluster 7 with high-level atrazine counties. The fate of atrazine in these two clusters could be due to the interplay between degradation processes, leaching to groundwater wells, and groundwater abstraction. In general, the expectation is that the greater the well depth or the depth to water table, the more the groundwater wells should be protected from atrazine contamination. Since these two clusters have similar well depth and groundwater temperature (based on annual mean air temperature), the difference in their mean atrazine concentrations may be due to differences in their precipitation, their rates of groundwater abstraction (as a function of the population of potential groundwater users and density of wells, Table 2), their bedrock geology, or whether they are in recharge or discharge areas.

Compared to cluster 5, the higher average groundwater atrazine concentration in cluster 7 could be attributed to leaching, since its counties are in recharge areas, coupled with higher precipitation and higher irrigation well density (more abstraction for irrigation during growing seasons). In contrast, the low average groundwater atrazine concentration in cluster 5 counties could be attributed to the fact that they are primarily in discharge areas (and close to major streams and rivers) with lower precipitation (less leaching), low irrigation well density, and high population (more drinking water wells). This could also result from groundwater mixing since some drinking water wells in discharge areas could draw water from nearby streams, lakes, or rivers.

The reasons for groundwater atrazine depletion are highly convoluted. This current study demonstrated robustness for deciphering the factors associated with low-level groundwater atrazine in counties of high atrazine usage. However, it was limited by reliance on publicly available data representing only county-level groundwater atrazine estimation. Atrazine measurement of abstracted groundwater at usage sites may be more explicit. Another limitation is that not all wells sampled for atrazine had measurements for atrazine metabolites. Hence, atrazine metabolites were under-reported in this study.

Furthermore, this study failed to account for the transport process of atrazine through the vadose zone. In addition, no data was available regarding atrazine at the recharge areas. However, county-level data summarized all the potential defects that the aforementioned limitations would provide if we had used individual wells for this study.

## 5. Conclusions

The motivation for this study emanated from our previous findings, which observed elevated ERC incidence in Nebraska counties with the highest atrazine usage. Given that groundwater is one of the significant exposure routes of atrazine to humans, we were puzzled by the low-level atrazine concentration frequently observed in the groundwater of these counties. As we unraveled the potential reasons for low-level groundwater atrazine in the counites of elevated ERC incidence and high atrazine usage, we found a negative correlation between well depth and groundwater atrazine, D.E.A., and D.I.A. concentrations. This suggests that shallow wells are more atrazine-contaminated than deeper wells. Further analysis among water supply wells with equal depths showed that excessive groundwater abstraction, reduced precipitation, high population, metropolitan areas, and water discharge areas were potential reasons for Nebraska’s observed low groundwater atrazine in high atrazine usage counties. Hence, this makes it difficult to rely on groundwater atrazine measurement as a good predictor for potential health implications of atrazine. Therefore, as we aim to determine the toxicity and health implications of atrazine in this field, groundwater atrazine may not sufficiently explain potential pathological implications; studies aimed at understanding the potential toxicity of atrazine in water should utilize atrazine measurement of already abstracted groundwater. They may be better predictors of health outcomes.

## Figures and Tables

**Figure 1 ijerph-18-13241-f001:**
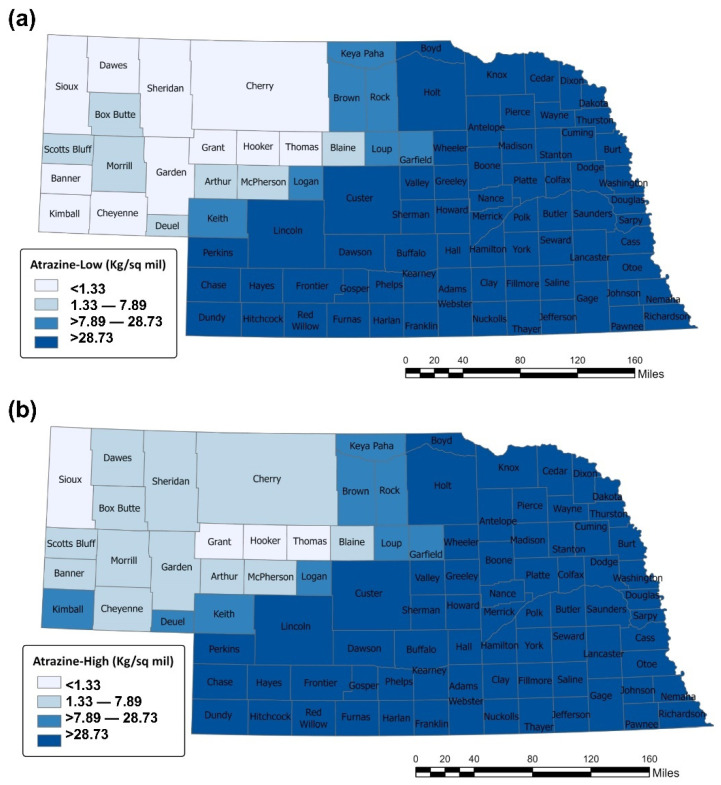
Estimated agricultural use for atrazine in Nebraska, National Water-Quality Assessment (NAWQA) Project, United States Geological Survey, 1995. (**a**) Atrazine (EPest-low) (**b**) Atrazine (EPest-High).

**Figure 2 ijerph-18-13241-f002:**
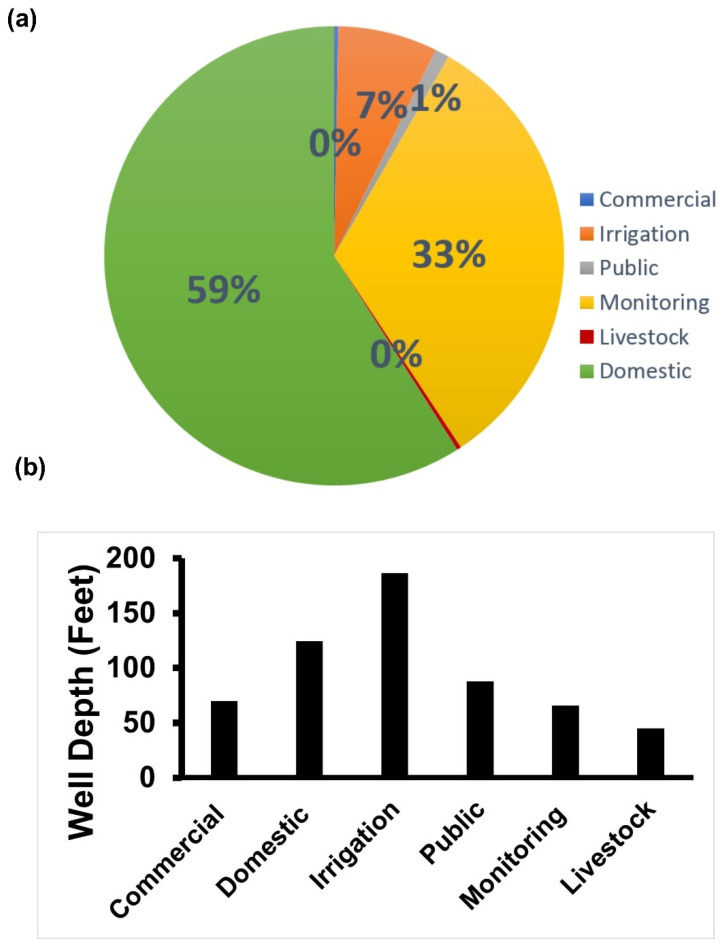
Descriptive characteristics of well types in the selected counties at the eastern district of Nebraska obtained from quality-assessed agrichemical contaminant Nebraska groundwater database. (**a**) The prevalence of well types (1995). (**b**) Average well depth of the different well types (1995).

**Figure 3 ijerph-18-13241-f003:**
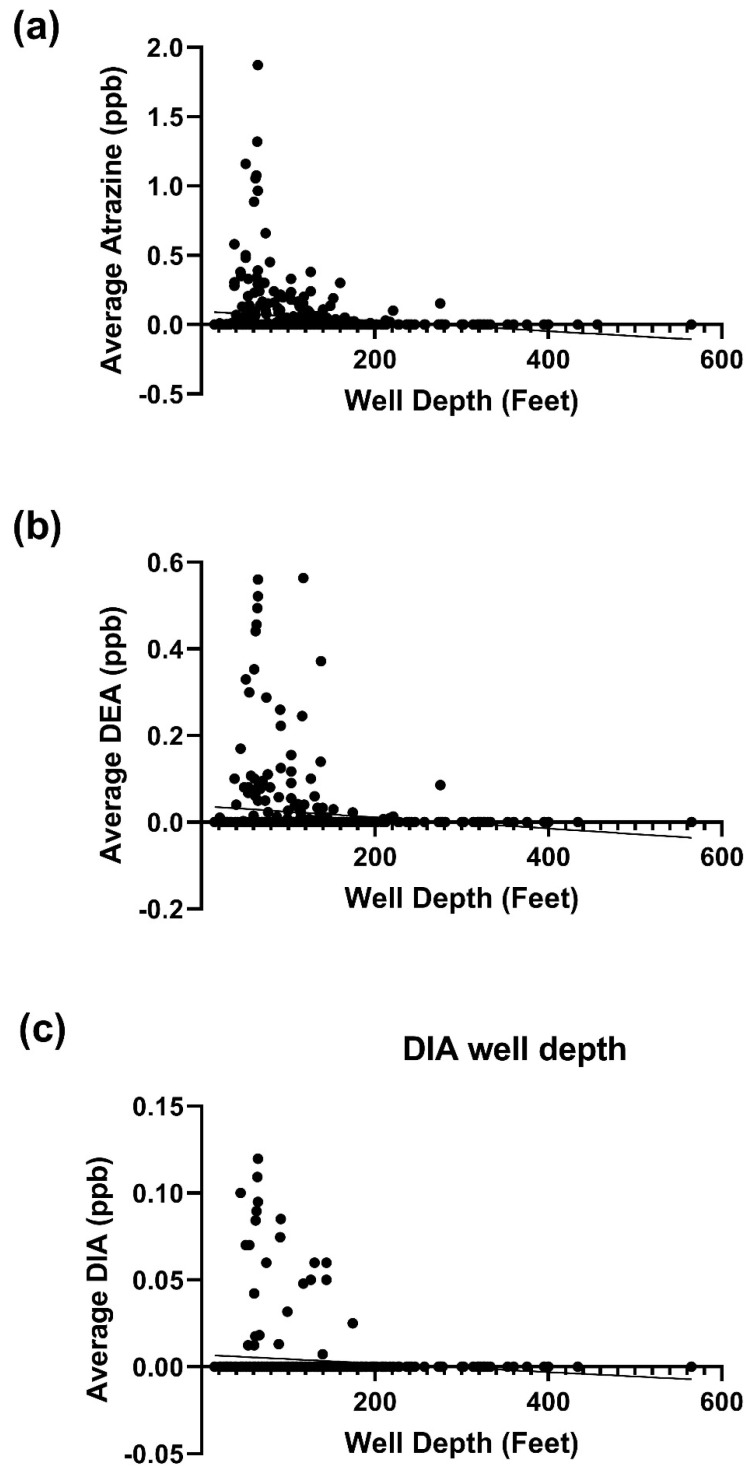
Atrazine and its metabolites based on well depths. (**a**) Average atrazine concentration detected in different well depths; (**b**) Average D.E.A. concentration detected in different well depths; (**c**) Average D.I.A. concentration detected in different well depth.

**Figure 4 ijerph-18-13241-f004:**
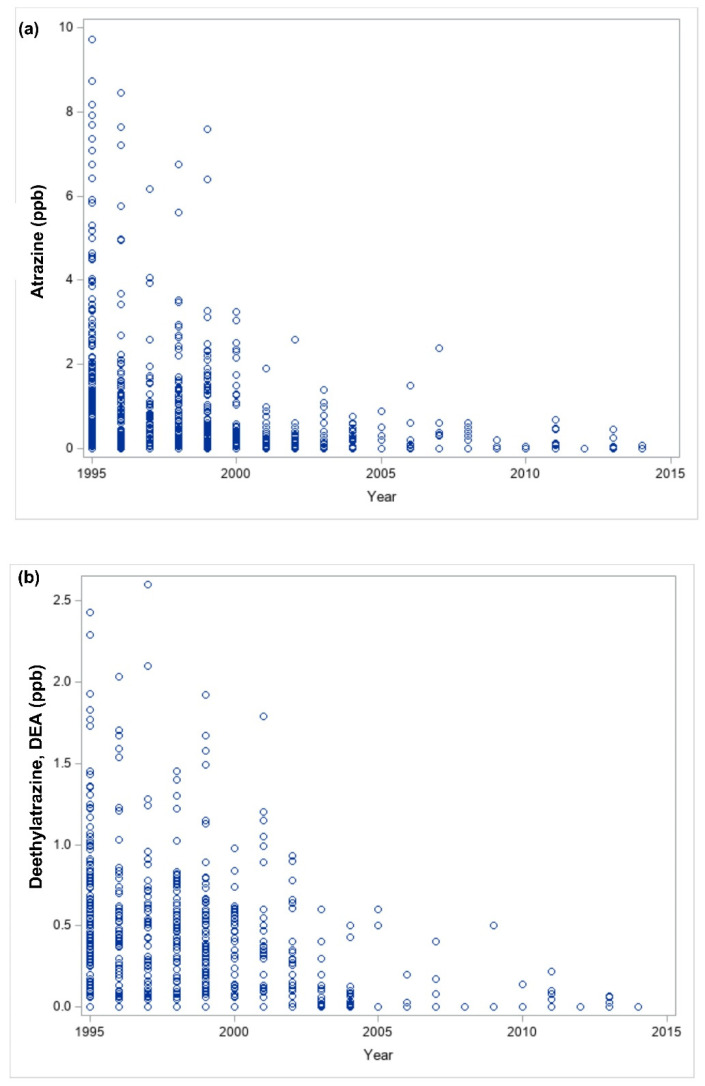
Linear relationship of atrazine and its metabolites over time (1995–2014). (**a**) Time series plot of atrazine (ppb). (**b**). Time series plot of D.E.A. (ppb). (**c**) Time series plot of D.I.A. (ppb).

**Figure 5 ijerph-18-13241-f005:**
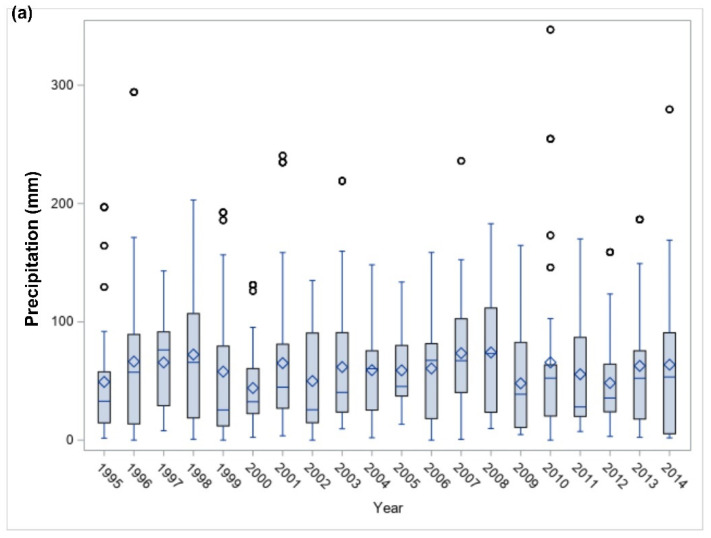
Climatic changes with time, PRISM (1995–2014). (**a**) Precipitation. (**b**) Temperature.

**Figure 6 ijerph-18-13241-f006:**
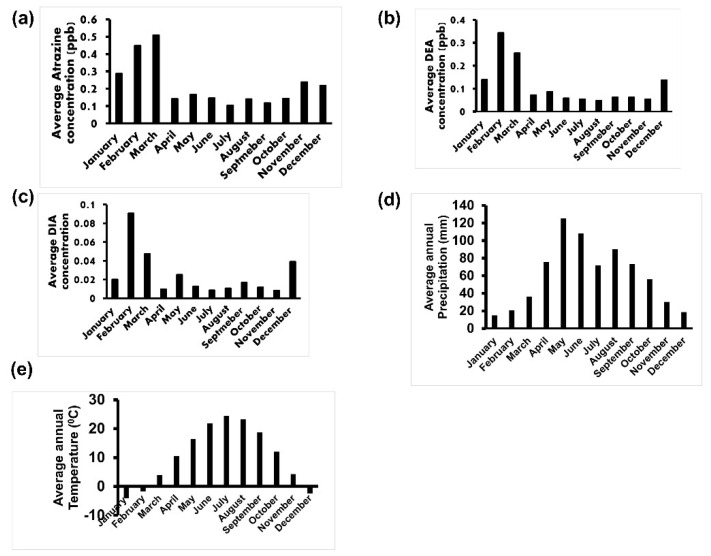
Seasonal variation in groundwater atrazine and its metabolites (**a**) Average atrazine concentration by month. (**b**) Average D.E.A. concentration by month. (**c**) Average D.I.A. concentration by month (**d**). Average annual precipitation by month. (**e**) Mean daily temperature by month.

**Figure 7 ijerph-18-13241-f007:**
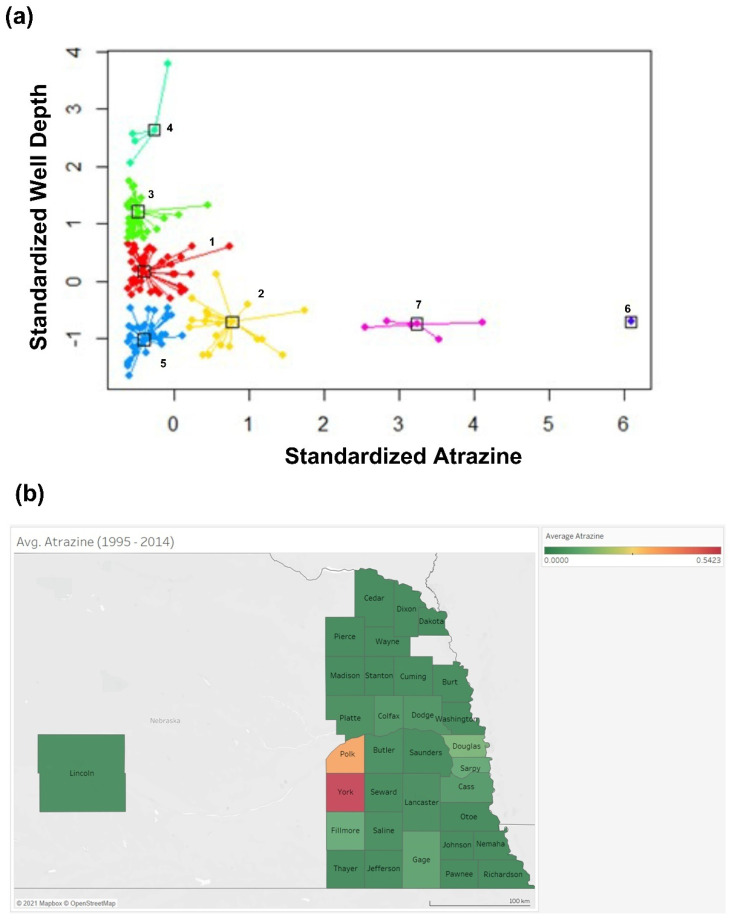
Cluster analysis to demonstrate the determinants of low-level groundwater atrazine, Quality Assessed Agrichemical Contaminant Nebraska Groundwater Database, 1995–2014. (**a**) Scatter plot of well depth versus atrazine groundwater concentration. Each color corresponds to a cluster, and a box marks each cluster’s prototypical data point (exemplar) while all cluster members are joined to their exemplars with lines. (**b**) Atrazine level by counties (**c**) Average precipitation by counties (**d**) Average temperature by counties (**e**) Population by counties (2018) (**f**) Changes in the groundwater level of counties of observed clusters (**g**) River basins of counties of observed clusters (**h**) Density of wells in the counties of observed clusters. Counties with utilization wells were labeled “U”, and counties with monitoring wells were labeled “M”.

**Table 1 ijerph-18-13241-t001:** The descriptive statistics of atrazine, its metabolites, and well depth obtained from quality-assessed agrichemical contaminant Nebraska groundwater database (1995–2014), precipitation and annual mean air temperature obtained from PRISM climate data (1995–2014), and amount of water usage obtained from water use data (2010) in the selected counties of eastern Nebraska district.

Variable	N	Mean	Standard Deviation	Minimum	Maximum
Atrazine (ppb)	4053	0.2	0.7	0.0	9.7
Deethylatrazine(ppb)	3008	0.02	0.08	0.0	0.8
Deisopropylatrazine (ppb)	3601	0.07	0.2	0.0	2.6
Hydroxyatrazine (ppb)	70	0.0	0.0	0.0	0.0
Well depth (feet)	4295	129.9	85.3	4.0	765.0
Precipitation (mm)	3360	60.2	53.2	0.0	347.0
Mean daily temperature (°C)	3360	10.6	10.3	-10.9	27.2
Domestic total self-supplied groundwater withdrawals (Mgal/d)	33	0.90	1.9	0.02	10.1
Domestic self-supplied population (thousands)	33	7.1	15.0	0.1	79.1

**Table 2 ijerph-18-13241-t002:** Comparisons of cluster 5 and cluster 7 characteristics.

Characteristics	Cluster 5	Cluster 7
Counties	Burt, Cass, Colfax, Dodge, Lincoln, and Sarpy	York and Polk
Number of observations	34	6
Average of precipitation (mm)	57.7	62.0
Annual average air temperature (°C)	10.8	10.8
Average well density (wells per area of land)	Low	High
Average domestic self-supplied population, in thousands	7.30	2.90
Average public supply population served by groundwater, in thousands	30.20	6.84
Average domestic total self-supplied withdrawals, groundwater, in Mgal/d	0.96	0.37
Average public supply total self-supplied withdrawals, groundwater, in Mgal/d	8.82	1.26
Average commercial total self-supplied withdrawals, groundwater, in Mgal/d	0.008	0
Average total population of the area (in thousands)	103.1	9.9
Metropolitan	Yes	No

## Data Availability

Please refer to https://clearinghouse.nebraska.gov/Clearinghouse.aspx, for Quality Assessed Agrichemical Contaminant Nebraska Groundwater Database, Omaha Nebraska, USA and https://waterdata.usgs.gov/ne/nwis/wu for Water Use Data obtained from United States Geographical Survey (USGS).

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
