# Peer review of "Low-Level Groundwater Atrazine in High Atrazine Usage Nebraska Counties: Likely Effects of Excessive Groundwater Abstraction"

_ijerph, 2021, doi:10.3390/ijerph182413241_

Round 1

Reviewer 1 Report

This is an interesting study. Author reported the reason why low-level groundwater atrazine concentration was observed in Nebraska counties. Interestingly, the ground water atrazine was affected by well depth, the one reason of low atrazine was characterized by excessive ground water abstraction. And also provided the evidence of seasonal variation of atrazine. In additional observed the correlation of atrazine depletion with the time. Overall, this is a good planned and performed study, and the conclusions are reasonable.  I think it is suitable for publication as submitted. I have no substantive criticisms. 

Author Response

Dear Reviewer, 

We appreciate your favorable reviews and remarks for our manuscript. We see you want our conclusion to be improved. We have improved our conclusion and we hope the current state of the manuscript is satisfactory for publication. 

Thank you. 

Reviewer 2 Report

Dear authors,

please take into account the following comments:

  • Figure 4 cannot be read properly. Please redraw.
  • Idem with Fig. 5
  • Figure 6: Please unify the way in which axis labels are oriented 
  • Figure 7: these figures are too small and letters cannot be read. A full redrawing of these pictures should be done so the reader can understand them.
  • The conclusions section is too short. A full new section must be written

Author Response

Dear Reviewer, 

Thank you for the thorough review of our manuscript. We have revised the manuscript as recommended and the revisions made were written in red fonts.

Figures 4,5,6 and 7 which were not clearly represented have been redone. The conclusion was rewritten to accommodate all the major findings of this study. 

We hope we were able to fully satisfy you during the review. 

Thank you very much.  

Author Response

Dear Reviewer, 

Thank you for the thorough review of our manuscript. In response to your review we have radically made changes to areas of concerns in the manuscript. Places where changes were made were written in red font. 

Please see below for the revisions made based on your comments and recommendations. 

Line 70-72: The toxicity of DEA and DIA have been described.

Line 106-107: The database has been capitalized (Line 111)

Line 111-112:  The selection criteria for the wells was highlighted on lines 88-96. Wells selected for this study were wells in eastern Nebraska counties with highest atrazine usage (Line 116-117). 

Line 130-137: We agree with the reviewer that other groundwater types would have similar effects on atrazine concentration as domestic groundwater. However, we are focused on demonstrating the effects of only water supply wells on groundwater atrazine depletion since domestic groundwater are better predictors of potential exposure of groundwater atrazine to humans. Moreover, we included other groundwater types to table 2 that may be important for human direct use. (Lines 138-145). 

Line 142: The reviewer is correct. We added “concentrations (ppb)” to atrazine, DEA and DIA.

Lines 145-146: Given the longitudinal nature of the data a pairwise correlation analysis was performed between atrazine, DEA and DIA and time. Moreover, a longitudinal data analysis was conducted but our findings did not explain groundwater atrazine usage. Hence, the longitudinal data analysis was excluded from this study.

Table 1: We have increased the space per the reviewer’s recommendation.

Fig 2a has been reworked per the reviewer’s recommendation. It must also be noted that pie chart approximates decimal percent to the nearest whole numbers. Hence, commercial and livestock groundwater with less than 1 percent was approximated to zero. That is why we do not see visible colors of commercial and livestock wells. Regarding figure 2b, well depths were expressed differently from figure 3 because well depth in figure 2b is a continuous variable while well depth in figure 3 were represented as categorical variables.

Figure 3: We agree with the reviewer. In response to that, figure 3a-c were changed  to scatter plot as recommended by the reviewer.

Figure 5 was a descriptive plot and may not require any statistical test to show significant differences for the precipitation and temperature.

Figure 6 has been formatted based on the reviewer’s recommendations.

In order to provide clarity to the reviewer’s concerns on lines 225-226, we have added an explanation to what is meant by discharge area (lines 237-252)

Table 2: “M” means monitoring wells, but it has been removed from table 2.

The reviewers concerns on discharge areas was addressed on Lines 242-256. 

For line 264, the reviewer’s recommendation was adopted (Lines 295-299)

Lines 334-335: We agree with the review that a more metropolitan area may mean lower agricultural activity and lesser atrazine use. However, this is not the case in Nebraska. Agriculture activities in the Metro area are as high as the rural settings.  The reason for identifying areas that are metropolitan was to emphasize the population density in the metro area which may justify higher groundwater abstraction by the increasing population.